# On the Feasibility of Adapting the LiVec Tactile Sensing Principle to Non-Planar Surfaces: A Thin, Flexible Tactile Sensor

**DOI:** 10.3390/s25082544

**Published:** 2025-04-17

**Authors:** Olivia Leslie, David Córdova Bulens, Stephen J. Redmond

**Affiliations:** School of Electrical and Electronic Engineering, University College Dublin (UCD), D04 V1W8 Dublin, Ireland; olivia.leslie@ucdconnect.ie (O.L.);

**Keywords:** tactile, optical, flexible

## Abstract

Tactile sensation across the whole hand, including the fingers and palm, is essential for manipulation and, therefore, is expected to be similarly useful for enabling dexterous robot manipulation. Tactile sensation would ideally be distributed (over large surface areas), have a high precision, and provide measurements in multiple axes, allowing for effective manipulation and interaction with objects of varying shapes, textures, friction, and compliance. Given the complex geometries and articulation of state-of-the-art robotic grippers and hands, they would benefit greatly from their surface being instrumented with a thin, curved, and/or flexible tactile sensor technology. However, the majority of current sensor technologies measure tactile information across a planar sensing surface or instrument-curved skin using relatively bulky camera-based approaches; proportionally in the literature, thin and flexible tactile sensor arrays are an under-explored topic. This paper, presents a thin, flexible, non-camera-based optical tactile sensor design as an investigation into the feasibility of adapting our novel LiVec sensing principle to curved and flexible surfaces. To implement the flexible sensor, flexible PCB technology is utilized in combination with other soft components. This proof-of-concept design eliminates rigid circuit boards, creating a sensor capable of providing localized 3D force and 3D displacement measurements across an array of sensing units in a small-thickness, non-camera-based optical tactile sensor skin covering a curved surface. The sensor consists of 16 sensing units arranged in a uniform 4 × 4 grid with an overall size of 30 mm × 30 mm × 7.2 mm in length, width, and depth, respectively. The sensor successfully estimated local XYZ forces and displacements in a curved configuration across all sixteen sensing units, the average force bias values ($\bar{\mu }$) were −1.04 mN, −0.32 mN, and −1.31 mN, and the average precision ($\bar{SD}$) was 54.49 mN, 55.16 mN and 97.15 mN, for the X, Y, Z axes, respectively, the average displacement bias values ($\bar{\mu }$) were 1.58 μm, 0.29 μm, and −1.99 μm, and the average precision values ($\bar{SD}$) were 221.61 μm, 247.74 μm, and 44.93 μm for the X, Y, and Z axes, respectively. This work provides crucial insights into the design and calibration of future curved LiVec sensors for robotic fingers and palms, making it highly suitable for enhancing dexterous robotic manipulation in complex, real-world environments.

## 1. Introduction

Mechanoreceptors are distributed throughout the human hand, with the highest concentration in the fingertips, enabling comprehensive tactile feedback, which is essential for dexterous manipulation [1,2]. The natural curvature of the human fingertip and sensation across the skin of the fingers and palm enhances object control by inherently ensuring consistent surface contact, aiding in collision detection and re-grasping [1,3,4]. The curved shape of the finger also allows for better conformability to the surface of objects, enabling more stable and implicitly adaptive grips [3]. It has been shown that when humans manipulate objects, there is significant use of the curved sides of the fingers [5,6]. This curvature facilitates even force distribution, improving the reliability of the grasp and minimizing the risk of slippage or damage to the object [3].

Replicating human-like tactile capabilities in robotic systems requires comprehensive sensing across the entire surface of the end-effector. These sensors must provide high-precision, localized measurements of multi-axis force and deformation, particularly on non-planar, curved surfaces such as robotic phalanges and palms. Tactile feedback in these areas is crucial for achieving a secure and stable grip. However, most existing tactile sensors capable of measuring multi-axis force and deformation are flat or planar [7,8,9,10]. This design limitation fails to replicate the curved nature of human fingers and restricts the amount of tactile data that can be collected.

Flexible tactile sensors offer a promising alternative, as they can conform to the non-planar surfaces of robotic fingers, phalanges, and palms [11]. However, the complex curvature of robotic hands presents significant technical challenges for sensor design. Developing flexible tactile sensors that provide multi-axis force sensing and detailed slip information requires balancing flexibility, performance, and comprehensive data acquisition [3,11]. As a result, many existing flexible sensors remain limited in their measurement capabilities. For instance, Ji et al. introduced a templated laser-induced graphene (TLIG)-based tactile sensor with high sensitivity, broad detection range, and excellent stability, making it well suited for applications like wearable health monitoring and texture recognition [12]. However, despite these strengths, the sensor is primarily limited to pressure measurement. Similarly, Yang et al. developed an iontronic pressure sensor using vertical graphene (VG) electrodes, achieving ultra-high sensitivity and mechanical stability alongside the ability to recognize different object shapes [13]. Another approach by Chen et al. utilized a biomimetic tactile sensing array with programmable microdome structures to enhance pressure sensitivity and spatial resolution [14]. While these methods are highly flexible and durable, they focus on pressure sensing rather than capturing full tactile interactions such as shear force or slip.

A common approach to flexible sensor design for robotic hands involves the use of flexible PCB technology, enabling tactile sensors to conform to curved surfaces and articulate at robotic joints [15,16]. For example, Wang et al. developed a flexible 3 × 3 array of capacitive elements capable of measuring contact force and slip, designed to wrap around the curved surface of a human thumb [17]. A later iteration extended this design to robotic fingers, allowing for joint articulation [18]. While this capacitive approach enables thin and flexible sensors, it remains limited in force range (maximum of 3 N) and is prone to mechanical coupling interference and cross-talk between sensing units [18]. Similarly, piezoresistive sensors have been explored for flexible tactile sensing, such as Buscher et al.’s pressure-sensitive glove, which wraps around a robotic gripper [19]. Zhu et al. proposed another approach with three smaller piezoresistive sensors distributed across a robotic phalanx, allowing the sensor to flex at the joints [20]. However, while highly flexible, piezoresistive sensors primarily measure pressure and lack full multi-axis force sensing capabilities [19,20,21].

Another notable example of a curved robotic fingertip sensor is the uSkin sensor, which employs a magnetic sensing principle with an array of permanent magnets suspended above Hall effect MEMS sensors [22]. Its small, modular design allows it to be effectively integrated into robotic hands such as Allegro and iCub. However, magnetic sensors like uSkin are inherently vulnerable to external magnetic interference from commonplace sources such as ferromagnetic materials, which can compromise sensing accuracy.

In contrast, optical sensing techniques offer several advantages, including high sensitivity, immunity to electromagnetic interference, and the ability to perform multi-axis, multi-measurement sensing. Camera-based sensors are capable of providing high-resolution measurements and often feature non-flat, curved skins. Many incorporate hemispherical tips to mimic the human fingertip [7,8,23,24,25,26,27,28], while others extend sensing capabilities along cylindrical finger surfaces [25,29]. Among these, the GelSight Fin Ray sensor shows promise due to its ability to passively adapt to robotic grippers, providing high spatial resolution, three-axis force sensing, and object orientation estimation [30,31]. This is achieved by embedding mirrors and a camera within a fin ray effect gripper. A recent development is Meta’s Digit 360, a compact optical sensor approximately the size of a human fingertip. It is capable of measuring three-axis forces in the milli-newton range while also detecting vibrations and spatial features with a resolution as fine as 7 microns [32]. This is an advancement in multi-directional camera-based optical sensing within a small, efficient form factor. However, its applicability remains constrained by its rigid structure, limiting its ability to conform to curved surfaces. Despite their advantages, camera-based sensors encounter several challenges in flexible sensing applications. Their inherent rigidity and bulkiness limits their ability to conform to highly curved or articulated surfaces.

Alternatively, optical fibers have been explored as a flexible optical tactile sensing solution due to their thin, lightweight, and highly adaptable nature [33]. Jiang et al. demonstrated the use of an optical microfiber-based slip sensor, approximately 2 mm thick, capable of measuring three-axis force. However, despite incorporating fingerprint-mimicking ridges, this sensor is limited to a single measurement point over a relatively large sensing area of 2.5 cm × 7.5 cm.

The advantages of optical sensing for multi-axis tactile perception, combined with the limitations of existing optical approaches, showed the need for non-camera-based optical tactile sensors capable of instrumenting complex non-planar surfaces, such as robotic palms and fingers. In our previous work, the LiVec Finger was presented [34], which offers a thin, distributed multi-axis measurement sensor, using our novel non-camera-based optical sensing principle referred to as the LiVec principle [34,35]. However, the current LiVec Finger design is limited to tactile sensing over a planar sensor array area, requiring the object surface to be oriented parallel to the sensor surface (or vice versa) to ensure both a secure grasp and adequate tactile sensation regarding the contact interface. To allow robotic grippers to work more effectively in unstructured environments, having curved fingers with tactile sensing can reduce the need for perfectly normal contact between the sensor and the object. Therefore, having tactile sensation on all parts of a robotic gripper (fingers and palm) is beneficial. However, implementing the LiVec sensing principle to such non-planar surfaces requires us to adapt the sensor design.

Therefore, this paper presents a novel innovation of the LiVec sensor array concept: the Flexible LiVec sensor. This proof-of-concept design uses a flexible PCB and eliminates the rigid circuit board, creating a sensor capable of providing localized 3D force and 3D displacement measurements across an array of sensing units in a small-thickness, non-camera-based optical tactile sensor but across a curved surface. The aim is that the Flexible LiVec sensor will ultimately be able to conform to curved surfaces, such as the palms or joints of robotic hands, delivering precise, distributed tactile sensation across the entire robotic hand and beyond. The design concept combines the advantages of flexibility and a dense array of sensing units in a compact form, intending to make it well suited for integration into the curved areas of existing robotic grippers.

The contribution of this paper is a demonstration of the feasibility of developing a thin, flexible, non-camera-based optical tactile sensor that can conform to curved surfaces. The sensor presented here is a novel iteration of previous designs in the LiVec sensing family, now using flexible PCB technology to instrument non-planar surfaces. The flexible sensor is characterized in multiple configurations, demonstrating its capability for real-time measurements of distributed 3D displacement and 3D force, providing crucial insights on future calibration requirements of designs.

The rest of this paper is organized as follows. Section 2 describes the design and fabrication of the flexible skin-like sensor using the LiVec sensing principle. Additionally, the experimental calibration procedure is presented, which includes two configuration variations for investigation. Section 3 presents the results for the two calibration configuration variations. Section 4 compares the flexible skin-like LiVec sensor against the sensing capability and precision of the previous LiVec sensors [34,35] and to various state-of-the-art flexible tactile sensors. Section 5 draws conclusions and suggests future work.

## 2. Materials and Methods

### 2.1. Design and Fabrication of the Flexible LiVec Sensor

The Flexible LiVec sensor consists of 16 sensing units arranged in a uniform 4 × 4 grid, shown in Figure 1. Each sensing unit can estimate 3D point forces and displacements. The assembled sensor has an overall size of 30 mm × 30 mm × 7.2 mm in length, width, and depth, respectively (Figure 1C,D). The 16 sensing units have a 7 mm pitch between units, a 6 mm external diameter at the base, and a maximum height of 5 mm. This design varies from the LiVec Finger presented in [34], shown in Figure 2, by having the following:A 14% decrease in center-to-center sensing unit density (7 mm vs. 8.15 mm).A 40% reduction in thickness (7.2 mm vs. 12 mm).A more compact design, containing 60% more sensing units (dimensions: 30 × 30 mm^2^ vs. 38.22 × 26.44 mm^2^).A removal of the rigid top plate and backbone assembly components to allow the sensor to be flexible.A flexible PCB.

The Flexible LiVec sensor uses the LiVec sensing principle [35]. Each local sensing unit consists of a deformable, cylindrical protrusion encompassing the local sensing components (see Figure 3). As each sensing unit is individual, this means that future designs can be made modular, allowing the arrangement of the sensing units to be specific to robotic phalanges or palms. The size of the flexible LiVec sensor is primarily constrained by the dimensions of its individual sensing units, which are determined by the off-the-shelf light-angle sensor and photometric front-end components. These factors influence both the overall sensor footprint and the density of sensing units in a uniform layout.

The LiVec sensing principle is presented in [34] in detail; the following is a brief summary. Each LiVec sensing unit infers XYZ force and displacement by measuring light reflected off an internal reflector within the skin protrusion. When an external force deforms the skin protrusion, this shifts the internal reflector’s position (and possibly orientation), altering the reflected light’s direction and intensity. The light angle sensor captures this light and outputs four photocurrents, encoding light ray angles of arrival and intensities. Each sensing unit undergoes a calibration procedure to convert the photocurrents into XYZ force and displacement estimates. The vector sum of the local XYZ forces across all sensing elements is used to estimate the global force of the whole sensor array.

#### 2.1.1. Electronics

The sensor electronics are manufactured on a flexible, two-sided, 6-layer PCB. The sensing components are mounted on the top side of the board, i.e., the light angle sensors and infrared LEDs. The four middle PCB layers are used for routing, and the fifth layer is reserved as a dedicated ground plane. The complete design of 16 pillars uses 32,850 nm wavelength infrared LEDs (IN-S42CTQIR, Inolux Corporation, Santa Clara, CA, USA) and 16 light angle sensors (ADPD2140, Analog Devices, Norwood, MA, USA), which are interfaced with eight photometric front ends on the back of the board (ADPD1080, Analog Devices). Each ADPD1080 photometric front-end chip is used to control LED pulsing and associated synchronized readout (using a 14-bit ADC) of two ADPD2140 light angle sensors; LED pulsing enables both ambient light rejection and power saving. A Teensy 4.1 microcontroller reads data from each ADPD1080 photometric front end using an I^2^C protocol whenever triggered by a data-ready interrupt and transmits these data over a serial connection to a PC.

#### 2.1.2. Skin Fabrication

The sensor skin was molded from a two-part platinum-cured silicone (Mold StarTM 20T, Smooth-On, Macungie, PA, USA) with a Shore hardness of 20A in a 1:1 ratio (Figure 4). The skin molds are 3D printed using stereolithography (FormLabs Form 2 printer, using Clear v4 resin, FormLabs, Boston, MA, USA). Once printed, the molds are post-processed by rinsing with isopropyl alcohol for 25 min and then curing at 60 °C for 15 min (Figure 4). The bottom molds form the external surface of the skin, and the top mold forms the internal cavities of the sensing units. To mold the skin, the silicone mix is dyed with a black pigment, degassed, then poured into the bottom mold. The upper mold is then attached and the mix left to cure. Once the black silicone skin has cured, the white reflectors are added to the internal cavity using a white-dyed silicone, using a pipette to add white silicone into holes (1.5 mm diameter and 0.3 mm depth) previously molded at the top of each interval cavity. The white external tracking dots (used later for position calibration) are again added to the external tip of each sensing element protrusion by pipetting white silicone into holes (0.5 mm diameter and 0.2 mm deep) previously molded into the external skin.

A base skin was fabricated to hold the PCB and to help affix the sensing unit skin to the PCB. The base skin was 3D printed on a Stratasys J35 Pro (Stratasys, Eden Prairie, MN, USA) in a soft rubber-like material known as ElasticoBlack. Custom-embedded guide holes were printed within the base skin to secure the metal screws.

#### 2.1.3. Mechanical Design and Assembly

The assembled Flexible LiVec sensor prototype consists of three parts: the sensing unit skin, the flexible PCB, and the base skin (Figure 3B); both of the skins are soft and flexible. Each internal reflector, located at the top of the truncated cylindrical internal cavity, has a diameter of 1.5 mm and is located at a height of 3.45 mm from the PCB. The sensor is 7.2 mm thick from the back of the base skin to the top of the sensing units. The construction varies from the LiVec sensor designs previously published [34,35] by removing the top plate, which was rigid, as a rigid top plate would restrict the flexibility of the sensor.

### 2.2. Local Force and Displacement Calibration Procedures

The Flexible LiVec sensor was calibrated in two configurations, flat (non-flexed) and flexed (curved), and the calibration equations were tested in three configurations to assess the effect of flexing the sensor. This was performed to evaluate if the Flexible LiVec sensor is still functional and accurate when flexed. Ideally, the Flexible LiVec sensor remains accurate when using the calibration coefficients from the flat configuration experiments.

The three testing variations are as follows: (1) training data for the calibration process collected in a flat sensor configuration, tested with data from the flat configuration (referred to as Flat–Flat); (2) training data for the calibration process collected in a flat sensor configuration, tested with data from the curved configuration (referred to as Flat–Curved); and (3) training data for the calibration process collected in a curved sensor configuration, tested with data from the curved configuration (referred to as Curved–Curved).

The calibration setup and procedure are identical to the previously designed test rig and procedure for previous versions of the LiVec sensors [34,35]. The calibration methods are included below for completeness to the reader and to allow for full understanding of the experimental procedure here.

#### 2.2.1. Calibration Platform

The Flexible LiVec tactile sensor prototype is mounted on top of a six-axis force/torque sensor (ATI Mini 40, ATI Industrial, Automation, Dallas, TX, USA), which is in turn mounted on top of a six-degree-of-freedom (DoF) hexapod robot stage (H-820, Physik Instrumente, Karlsruhe, Germany). To collect calibration data for any sensing unit in the array, the robotic stage brings that sensing unit into contact with a 4 mm × 4 mm cubic outcrop (made of transparent acrylic) in order to apply a force to it and deform the tactile sensing unit. A video camera (Logitech Streamcam, Logitech, Lausanne, Switzerland; resolution of 1280 × 720 px; 60 FPS) is mounted 250 mm above the sensor and views the XY sensing unit external tracking dot position in the world frame. This setup is illustrated in Figure 5A,C. Force/torque data were sampled at 200 Hz, and all Flexible LiVec sensor data were sampled at 213 Hz. All data were later uniformly resampled offline to match the video sampling rate of 60 Hz.

#### 2.2.2. Calibration Procedure

The calibration procedure (briefly summarized in Figure 5B) was identical to that used for previous sensor designs in the LiVec sensor family [34,35]. Briefly, the robotic stage follows spiral and spoke patterns at varying compressions (Z displacements) ranging from 0.00 mm to −1.50 mm in −0.10 mm increments. The robotic stage traces these trajectories at a speed of 2 mm·s^−1^. The position encoder of the robotic stage and the XY position estimates from the camera tracking are used to estimate the displacement of the external tracking dot on the sensing element in the frame of reference of the LiVec sensor. The Z position of the robotic stage, from an offset position defined when first contact is made between the tactile sensor and the acrylic outcrop, defines the Z displacement of the sensing units in the Flexible LiVec frame of reference.

Independent training and validation datasets were acquired with this procedure for each of the sixteen sensing units.

Subsequent to data collection, for each of the 16 sensing units, 6 multivariate 4th-order polynomial regression models were trained to estimate the force and displacement vector experienced at the sensing unit tracking dot position; see Leslie et al. [34] for details. A light intensity variable (sum of the four biased photocurrent signals) is calculated using 200 samples of data when the sensing unit is known to be in an undeformed neutral position. This intensity variable is used to normalize (by division) all future photocurrent readings for that sensing element. The four normalized photocurrents and the intensity variable form the five input variables to the regression models for that sensing element, and this is similar for other sensing elements.

Precision (i.e., sample standard deviation) and bias (i.e., sample mean) statistics of the model estimation errors were used as measures of model accuracy.

##### Configurations Tested

To evaluate the Flexible LiVec sensor in the unflexed position, the sensor is mounted on a flat surface on the calibration platform described above (Section 2.2.1 and Figure 5C), and the calibration procedure is performed for each sensing unit. Then, a second run of the calibration procedure is performed to obtain a separate test dataset for each sensing unit.

To evaluate the Flexible LiVec sensor in a flexed position, the sensor was bent around the arc of an 8 cm radius circle and attached to the calibration platform used in Section 2.2.1 (see Figure 6). The bend radius of 8 cm was chosen as the maximum curvature of the Flexible LiVec sensor, as this is what the sensor could bend to with ease without risking the sensor breaking. The calibration procedure is described above (Section 2.2.2), i.e., spiral and spoke movements at various Z compressions (0.0 mm to −1.5 mm in −0.1 mm increments), and was performed for each sensing unit. Separate training and testing datasets were acquired for each of the sixteen sensing units in this flexed configuration. This flexed position was chosen to test the performance of the sensor when it is adapted to the curve of a robotic finger or palm and, in this case, a large curvature.

##### Assessment of Calibrations

To assess the accuracy and precision of the calibration equations in the various configurations, the force and displacement estimates of each sensing unit were compared using the calibration equations of a given configuration against the reference (“truth”) force and displacement values of one of the test datasets.

To assess how the sensor performs in the unflexed position, the calibration equations obtained with the sensor in a unflexed configuration were tested with the test dataset from the unflexed configuration, and these are referred to as Flat–Flat.

To assess the accuracy and precision of the calibration equations obtained with the sensor in a flat configuration when the sensor was flexed, the outputs obtained using the flat calibration equations were measured against the test dataset in a flexed configuration. This condition is referred to as Flat–Curved. This was conducted to assess how well the calibration performed when flexed to an unknown curvature, i.e., the palm of a robotic hand.

Finally, the accuracy and precision of the calibration equations obtained with the sensor in a flexed configuration were tested using the test dataset obtained with the sensor in a flexed configuration. This condition is referred to as Curved–Curved. This was performed to determine whether calibrating the sensing units when the sensor is in its flexed configuration (bent over the 8 cm radius arc) yields more accurate force and displacement estimates, i.e., when the sensor is in a permanently flexed position.

## 3. Results

Each calibration variation showed the Flexible LiVec sensor could successfully estimate local XYZ force and displacement for all sixteen sensing units. This shows that using a flexible PCB still allows the sensor to function well.

### 3.1. Three-Dimensional Force and 3D Displacement Validation of Each Sensing Unit

#### 3.1.1. Flat Configuration Sensing Validation

The obtained regression equations in the flat configuration were validated in the flat configuration using the flat test dataset (referred to as Flat–Flat). Estimated and reference displacements for each sensing unit matched well (see the example for sensing unit 8 in Figure 7A), and this was similar for force estimates when referenced to the force/torque sensor data (see the example for sensing unit 8 in Figure 7B).

The average (across all sixteen sensing units) force and displacement error statistics for all three tested configurations are visualized as error bars in Figure 8 and summarized in Table 1. These error statistics are similar to the LiVec Finger [34]. However, the LiVec Finger had more consistent force precision, with less variation between the axes. The precision of the force estimates for the previous LiVec Finger design differed between the X and Y axes by 1.70 mN (X: 20.89 mN, Y: 19.19 mN), while the precision of the force estimates from the Flexible LiVec sensor presented here differed between the X and Y axes by 28.57 mN. Furthermore, the LiVec Finger force biases had a narrower range, with a difference of 1.30 mN between the X and Y axes compared to a 2.95 mN difference of the Flexible LiVec sensor. Similarly, the LiVec Finger had more consistent and precise displacement estimates than the Flexible LiVec sensor. The LiVec Finger displacement precision in the X and Y axes was 56.70 μm and 50.18 μm, respectively, whereas the Flexible LiVec precision values are larger and with a bigger difference between the X and Y axes (180.89 μm and 172.26 μm, X and Y axes, respectively). Therefore, the Flexible LiVec sensor displacement precision is approximately three times worse than the LiVec Finger displacement precision.

The errors across the sensing unit force and displacement estimates when tested in the Flat–Flat configuration are centered around or near zero, and all tended to be relatively symmetric (Figure 9). The force error distributions are consistent across all the sensing units. The displacement error distributions vary more between sensing units, with a larger spread (Figure 10).

#### 3.1.2. Flexed Configurations Sensing Validation

To assess the effect of the flex configuration on the performance of the Flexible LiVec sensor, the sensor was mounted in a flexed configuration on a cylindrical fixture with a radius of 8 cm, and the XY calibration patterns (spiral and spoke) were applied on each sensing unit at different Z compressions. In this configuration, two situations were tested: (1) the calibration equations obtained with the training dataset collected in a flat sensor configuration, tested with the test dataset obtained in the curved configuration (referred to as Flat–Curved), and (2) the calibration equations obtained using the training dataset collected in a curved sensor configuration, tested with the test dataset obtained in the curved configuration (referred to as Curved–Curved).

Again, the average (across all sixteen sensing units) force and displacement error statistics for these curved testing configurations are visualized as error bars in Figure 8 and summarized in Table 1. When the Flexible LiVec sensor was tested in the Flat–Curved configuration, across the 16 sensing units, average precision ($\bar{SD}$) increased in the X, Y, and Z directions by 348%, 152%, and 45%, respectively, relative to the Flat–Flat configuration (i.e., when both calibration and test configurations are the same for the Flat–Flat configuration). Moreover, given the small bias errors of 6.93 mN and −6.65 mN in the X and Y axes, respectively, the relatively large −171 mN bias error in the Z axis (see Figure 8B and Table 1) after flexing the calibrated sensor around the 8 cm curvature is notable. Interestingly, while the Z axis bias saw a large increase, the precision was only, on average, 70% worse than the precision measured in a flat testing configuration. This can be observed in Figure 7D, where the estimates of unit 8 follow the reference force/torque sensor but not as closely as in previous cases (i.e., Figure 7B).

Similar behavior was observed with the displacement estimates tested in the Flat–Curved configuration, with displacement estimate precision values increasing by 148.8% in X, 177.4% in Y, and 137.5% in Z. This is shown in Figure 8A, where the displacement estimates follow the reference pattern but are offset (large bias), and is also seen in Figure 7C, illustrating the wide precision and offset bias for the displacement estimates here.

When the Flexible LiVec sensor is tested in the Curved–Curved configuration, the bias and precision of the sensing unit force and displacement estimates improve compared to the Flat–Curved configuration. Since this configuration tests the sensor when it has the same curvature at which it was recalibrated, the bias for the sensing units’ force and displacement estimates are again close to zero, similar to the Flat–Flat configuration. Changes in the axes’ precision values are more complicated, with the precision of the X and Y axes worsening by 22.5% and 43.8%, respectively, relative to the Flat–Flat configuration but the Z axis precision improving from 109 μm to 44.93 μm. Figure 8 compares these results in error bars for all three calibration/test configurations, highlighting how calibration in the intended use configuration (i.e., intended curvature) improves the results. This is also clearly visible in Figure 7E,F, where the force and displacement estimates more accurately follow the reference values.

## 4. Discussion

This paper introduces a significant advancement of the LiVec tactile sensor array concept: a flexible tactile sensor array using a non-camera-based optical sensing principle (LiVec). The Flexible LiVec sensor can estimate local XYZ force and displacements across a densely arranged 4 × 4 sensing element grid.

Using a flexible PCB instead of a rigid PCB and removing the rigid base and top plate allow the use of the LiVec sensing principle to implement a flexible tactile sensor array. When calibrated and tested in a flat (unflexed) configuration, this Flexible LiVec sensor has similar measurement estimate errors to that of current state-of-the-art sensors (Table 2) [22,36,37,38,39,40]. Each sensing unit within the array has a displacement range of ±1 mm in the X and Y axes and −2 to 0 mm in the Z axis and a force measurement range of ±1 N in the X and Y axes and −2.2 to 0 N in the Z axis, with a maximum measurable global force magnitude in the Z axis of around 35 N.

The Flexible LiVec sensor improves miniaturization over other sensors in the LiVec sensor family, offering a thinner design and more densely packed sensing units that rival the most compact non-camera-based optical sensors available [10,36,41]. The Flexible LiVec sensor is 40% thinner than the LiVec finger, with an overall thickness from the tip of the sensing units to the back of the sensor casing of 7.2 mm. The compact arrangement of the sensing units in a uniform 4 × 4 grid design provides a higher spatial resolution than the LiVec Finger, with the shortest center-to-center distance between sensing units being 7 mm. Table 2 compares the characteristics of flexible tactile sensors, highlighting the wide measurement range and good precision of the flexible LiVec sensor. Table 2 also provides insight into the variety of shapes seen in flexible sensors designed for robotic fingers, typically specialized designs for a finger or a palm. The individual sensing unit design of the flexible LiVec means it can be designed in a modular configuration to instrument various robotic finger shapes.

### 4.1. Flexibility

Tactile sensation in human hands, particularly the fingers, is crucial for fine manipulation [42]. Having tactile sensation across the whole curved surface of the finger skin allows humans to control their grip more easily. Many tactile sensors do not cover surfaces of varying curvatures, such as a fingertip or finger phalanx [3], as they are flat or have defined hemispherical skin shapes. For instance, camera-based sensors are often hemispherical [7,8] or cylindrical [25,29], but they cannot easily be made to be flexible, limiting the types of surfaces they can cover.

Electronic skins are highly flexible and extremely thin, representing a significant area of research within flexible tactile sensors. They are designed to cover large surfaces, conforming to various curved shapes and providing sensation across palms, phalanges, and extensive areas of robotic hands and bodies. However, these sensors generally detect only a single property, such as pressure or shear forces [43,44,45]. This limitation is a significant drawback for robotic manipulation applications, which require distributed multi-axis force sensing [2,42]. Moreover, achieving such thin and flexible profiles necessitates specialized and costly manufacturing processes, contributing to their limited adoption.

The Flexible LiVec sensor in its current form factor (4 × 4 grid) is a trade-off between measurement capabilities and flexibility. In its current form, it cannot cover an arbitrary curved surface with multiple bends like the above-mentioned sensors (Table 2: sensors 5, 6, 7, 8, and 10). The PCB of the Flexible LiVec sensor limits the overall flexibility. Flexing beyond a radius of curvature of 8 cm required a force that felt excessive when applied by hand. As a result, to avoid breaking the sensor, further flexing beyond this limit was not attempted due to the risk of damage. Some capacitive sensors can achieve a nearly 80-degree bend [15]; further, there is a resistive sensor [16] whose bend radius is 3 times larger than our flexible LiVec sensor. A six-layer flexible PCB was used to implement the sensor. This ensured proper electronic functionality by providing a full ground plane and separating digital and analog signals, reducing the overall signal-to-noise ratio. Although the flexible PCB is around 85% thinner than the previous rigid version, it remains relatively stiff. This stiffness may result from the six-layer construction and the densely packed rigid components, including light angle sensors and LEDs on the top side and the photometric front end on the backside. To enhance the sensor’s flexibility, a circuit redesign exploring the possibility of relocating the photometric front-end chips farther away from the light angle sensors and thus removing both a number of electronic components and their associated wires (which may ease the routing requirements and require fewer PCB layers), could be a way to improve the PCB flexibility. However, a significant benefit of the LiVec sensing principle compared with the above-mentioned flexible skins is that it enables multiple multi-axis measurements in a distributed array, is sensitive to mechanical vibrations, and is robust against electromagnetic interference.

### 4.2. Sensing Range

Among flexible tactile sensors used in robotic grippers, the LiVec sensor stands out regarding measurement range (see Table 2). The Flexible LiVec sensor has a force measuring span of 35 N in the Z axis, which is significantly larger than similar flexible sensors. For instance, this span is ten times larger than the span of the sensor proposed in [18] and twice as large as the span of the flexible sensor in [21]. Moreover, the span of force measurement of the LiVec sensor is dictated by the Shore hardness of the silicone used and can, therefore, be modified if necessary.

### 4.3. Thickness

An advantage of our optical Flexible LiVec tactile sensor is its small thickness, which may allow it to be integrated with robotic hands, particularly on the palms or phalanges. The development of this novel LiVec sensor design was motivated by the need to reduce the thickness of optical tactile sensors, making them more easily integrated with existing robotic grippers. The Flexible LiVec sensor, with a thickness of 7.2 mm, is the thinnest version of the LiVec sensors and stands out as competitive with the current state-of-the-art tactile sensors (Table 2). Traditionally, optical tactile sensors have been bulky, but recent advancements have led to miniaturized versions, some approaching the size of human fingertips [23,25,46,47,48]. For example, the GelTip sensor is finger-shaped, with a diameter of 15 mm [25], and the GelSlim 3.0 has a thickness of 20 mm [49], which is a large reduction in size from the 60 mm thickness of the earlier versions of the GelSight sensors [8]. The GelSight Baby Fin Ray sensor, which integrates flexible mirrors to allow the fingertip to flex, has a thickness of 18 mm [31]. However, these smaller designs often rely on mirrors and wide-angle lenses, which can reduce sensing accuracy [46]. An alternative approach to size reduction is seen in the DIGIT Pinki, which uses a 15 mm diameter distal sensing element with proximal illumination and imaging to create a compact sensing unit attached to robotic fingers [50]. However, it has a maximum sample rate of only 30 Hz and may require careful integration with a robotic hand so that the proximal components and wires do not interfere with the robotic hand. A recent advancement is the Digit 360 from Meta, which is about the size tip of a human finger and can provide three-axis force in the milli-newton range sensing alongside vibration and spatial details as fine as 7 microns [32]. Additionally, it has an AI neural network for a fast response time. This is a big step forward for multi-directional sensing in a small, compact form factor for an optical sensor. However, it is still limited in the curvature of the surfaces it can instrument, as it is not flexible. The magnetic triaxial uSkin tactile sensor has a thickness of 6.05 mm, with a spatial density of 4.7 mm between individual elements [22]. Although this sensor is thinner and has a higher spatial density than the Flexible LiVec sensor, it relies on magnetic transduction methods, making it prone to electromagnetic interference. Additionally, the optical vision-based tactile sensor proposed by Zhang et al., which uses a bionic compound eye structure, is thinner than our sensor, with a thickness of 5 mm. However, this sensor cannot be made flexible due to the limitations imposed by using a CMOS array in its base [51].

### 4.4. Measurement Bias and Precision

#### 4.4.1. Flat Testing

The Flexible LiVec sensor was tested in a flat (unflexed) position and showed a reduction in precision compared to the LiVec Finger [34]. The average force precision of the LiVec Finger (Table 1) is better across all axes, with average precision values of 20.89 mN, 19.19 mN, and 43.22 mN for the X, Y, and Z axes, respectively. In comparison, the Flexible LiVec sensor exhibits higher precision values of 24.99 mN, 53.56 mN, and 135.33 mN for the same axes (Table 1). Similarly, the displacement precision of the LiVec Finger is more consistent, with values of 56.70 μm, 50.18 μm, and 13.83 μm for the X, Y, and Z axes. The Flexible LiVec sensor, however, shows larger displacement average precision values, particularly in the X and Y axes, with 180.89 μm, and 172.26 μm, respectively. Similarly, the average displacement precision of the LiVec Finger is more consistent, at values of 56.70 μm, 50.18 μm and 13.83 μm for the X, Y, and Z axes. The Flexible LiVec sensor, however, shows larger displacement precision values, particularly in the X and Y axes, with 180.89 μm and 172.26 μm, respectively. Although the bias is slightly smaller in some axes for the flexible sensor design, the overall precision of the LiVec Finger remains more reliable, particularly for applications requiring stable and accurate force and displacement measurements.

These differences in bias and precision could be attributed to design variations between the sensors. The Flexible LiVec sensor’s soft, flexible back contrasts with the rigid back of the LiVec Finger, which likely provides less stability to the sensing elements and contributes to a less consistent zero point, especially in the Z axis. Additionally, the absence of a top plate in the flexible design may allow more skin movement, leading to inaccuracies due to mechanical coupling from adjacent sensing units. In the previous LiVec sensor design, the top plate securely attached the deformable skin to the PCB, preventing blistering and ensuring the skin remained in place. In the flexible sensor, it was observed that compressing one sensing unit, particularly in the middle of the array, could cause slight movement in adjacent units, decreasing the sensor’s overall precision. The number of sensing units in the Flexible LiVec sensor was increased from 10 to 16; however, this change did not impact performance, as it was determined that the removal of the top plate to allow for flexibility in the design introduced the blistering, which was the cause of the negatively impacted precision.

#### 4.4.2. Curved Testing

To obtain accurate estimates of local force and displacement, calibration is required; however, for the Flexible LiVec sensor, the position of the sensor, either flat (unflexed) or curved (flexed), influenced the calibration outputs. When the sensor was calibrated in a flat configuration but later tested in a curved configuration, significant biases in X and Y measurements for both displacement and force were observed (see Figure 7C,D).

The observed biases can likely be attributed to the shift in the reflector position relative to the light angle sensor when moving from a flat to a curved configuration and/or the light angle sensor being angled instead of normal to the reflector. Calibrating the sensor while it is in the curved configuration helps to account for these positional changes, thereby improving measurement accuracy. The sensor was configured with a larger curvature to assess this performance, which caused this bias shift. Therefore, flat calibration might perform better at smaller curvatures, but calibration in the actual configuration in which the sensor is deployed becomes crucial as the curvature increases.

### 4.5. Limitations and Future Considerations

The aim of the Flexible LiVec sensor was to further develop the LiVec sensor family by adapting the sensing principle to curved or flexible surfaces of robotic hands, such as the curved phalange or palm. However, it is important to note some limitations in the design and testing of the flexible LiVec sensor.

One limitation is that the current design evaluation of the Flexible LiVec sensor was only calibrated and then tested in three curvature configurations (Flat–Flat, Flat–Curved, and Curved–Curved). Expanding testing to include multiple curvatures would provide a more comprehensive understanding of the sensor’s performance across a broader range of real-world applications. However, additional curvature testing would not change the conclusion that our proposed calibration method, while offering significant benefits, is not robust to post-calibration changes in curvature. The results show that calibration coefficients determined in a flat (unflexed) position exhibit significant biases when applied to data acquired in a curved/flexed configuration. However, these biases are reduced when the sensor is both calibrated and tested in the same curved/flexed configuration. This highlights the importance of calibrating the sensor in its intended flexed configuration prior to use and maintaining that curvature for a valid calibration.

The observed changes in calibration coefficients with varying curvature can be attributed to several factors. As the sensor flexes, the relative position of the reflector to the light angle sensor may shift, altering the detected optical signal. Additionally, the light angle sensor itself may become angled relative to the reflector, further affecting the measured light path. The shape of the internal cavity within the skin could also deform when the sensor is flexed, and the reflector may stretch, subtly changing its shape, all affecting the transmission, reflection, and reception of light relative to that observed in the curvature configuration for which the sensor was calibrated. All of these factors influence the optical path between the LEDs and light angle sensor, introducing systematic errors in the calibration coefficients, resulting in observed measurement biases.

A potential solution to mitigate these biases is the implementation of dynamic recalibration or compensation techniques to maintain sensor accuracy. This could be achieved through real-time correction methods, such as adaptive calibration algorithms that update the calibration coefficients based on detected changes in sensor flexion. Alternatively, model-based compensation techniques could be employed to estimate and correct the expected distortions caused by flexion. However, any such method must be designed for real-time implementation, ensuring minimal computational delay to prevent performance lag. Future work should explore these approaches to improve sensor robustness under dynamic flexion and enhance its performance in robotic applications.

Beyond adapting the calibration coefficients, material and structural optimization could further enhance sensor performance. One solution could be adjusting the stiffness of the sensor’s skin, which would allow for a trade-off between force measurement sensitivity and measurement range. Softer materials could enhance sensitivity to small force changes, whereas stiffer materials would enable a broader force measurement range by resisting deformation. A different solution could mount the flexible sensor onto a curved but rigid surface to help mechanically stabilize the sensing units, minimizing unwanted deformation of the skin that could otherwise introduce variability in sensor readings. Additionally, directly glueing the sensor skin to the PCB could reduce the risk of blistering, improving the precision of the sensor. However, in line with the conclusions of the paper, methods that aim to constrain the curvature of the sensor will require that the sensor is calibrated in this configuration prior to use.

The sensitivity of the sensor is also influenced by its physical size and layout, primarily due to the constraints imposed by the off-the-shelf light angle sensor. The current square grid layout of the sensing units is approaching the maximum achievable density given the current design approach. To reduce the overall sensor size while maintaining performance, all component spatial dimensions would need to scale proportionally to ensure that the sensitivity is preserved. However, this would subsequently require enhancement of the sensitivity of the detector or an increase in the intensity of the LEDs to retain sufficient signal-to-noise ratio.

Another limitation is that the current evaluation did not assess the sensor’s durability or the stability of its calibration coefficients during continuous dynamic applications. While the previous version of the sensor (LiVec Finger [34]) was successfully demonstrated in a basic robotic manipulation task, the Flexible LiVec sensor has not yet been tested in robotic applications but would be expected to perform similarly.

Previous work on the LiVec sensing principle [35] identified temperature-induced output drift as a limitation affecting measurement precision. However, this issue can be mitigated using a standard approach of regular rebiasing, a method commonly employed in other force/torque sensors [52,53,54,55]. Future work will focus on characterizing the behavior of the LiVec sensing principle more generally in response to varying environmental conditions, including changes in temperature and humidity.

Despite the limitations of the Flexible LiVec sensor, it still demonstrates valuable multi-axis sensing capabilities, which might be sufficiently accurate for less precision-critical areas of a robotic hand, such as the palm or a phalanx.

## 5. Conclusions

This work proposed a flexible skin-like tactile sensor array as a new member of the LiVec sensing family to investigate the feasibility of adapting the LiVec sensing principle to curved surfaces such as robotic palms and finger phalanges. This work demonstrated that the non-camera-based LiVec sensing principle can be instrumented using a flexible PCB. The design consisted of a 4 × 4 grid of distributed sensing units. Compared with previous LiVec sensors, this flexible sensor offers a fully soft, flexible design with a 40% decrease in thickness [34]. The thin thickness and flexibility of the sensor allow it to be integrated onto curved surfaces of the robotic hand, providing multi-axis force, displacement, and vibration sensing. This has led to the following main contributions:A thin, flexible, non-camera-based optical tactile sensor that can conform to curved surfaces;Crucial insights on how to design and calibrate future iterations of curved LiVec sensors for robotic fingers and palms.

Future work will explore the development of the sensor to fit the complex shape of a fingertip. While this flexible design shows great promise for the range of non-uniform surfaces it can cover, providing comprehensive tactile information, a restriction is the lack of multi-directional flexibility. Our future work will involve iterations covering specific robotic fingers and palms, providing a large non-uniform/planar sensing area alongside showing the utility of the sensing in robotic manipulation tasks.

## 6. Patents

S.J. Redmond, D. Córdova Bulens, O. Leslie, P. Martinez Ulloa. Optical Sensor Device. UK Patent EP4526638A1 [56].

## Figures and Tables

**Figure 1 sensors-25-02544-f001:**
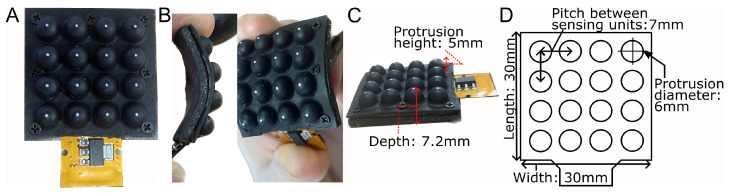
The fabricated flexible sensor prototype. (**A**) A front-view image of the sensor. (**B**) Images of the sensor in flexed positions. (**C**) A side-view image of the sensor showing the depth and sensing unit protrusion height dimensions. (**D**) A schematic drawing of the sensor with dimensions.

**Figure 2 sensors-25-02544-f002:**
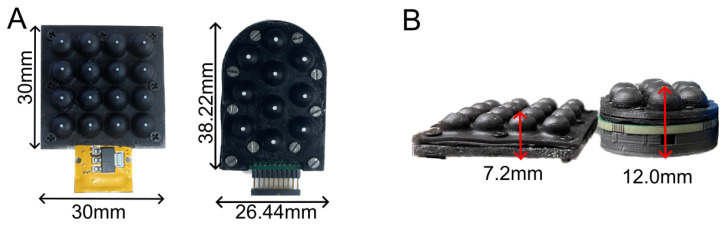
(**A**) Front-view images of the flexible sensor presented in this paper and the LiVec Finger presented in [34], showing the relative size differences, particularly how the flexible design spatial density of sensing units increases. (**B**) Side-view images of the flexible sensor and the LiVec Finger, showing the reduction in depth the flexible sensor achieves.

**Figure 3 sensors-25-02544-f003:**
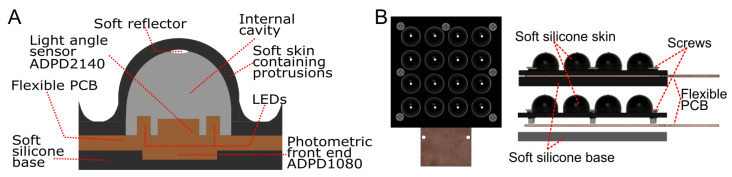
(**A**) A diagram of a sensing unit of the flexible sensor, showing the components. (**B**) CAD illustrations of the flexible sensor, showing the layering of the sensor components.

**Figure 4 sensors-25-02544-f004:**
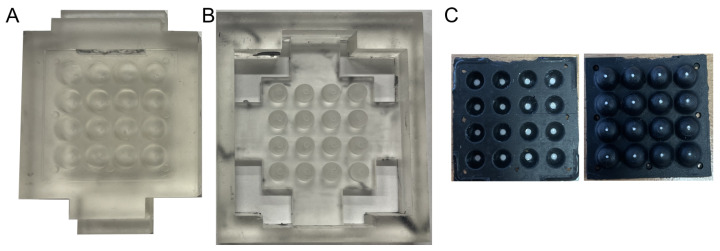
The molds for fabricating the flexible sensor skin. (**A**) The bottom mold. (**B**) The top mold. (**C**) The fabricated skin. The left image shows the inside of the skin and the right image shows the outside of the skin.

**Figure 5 sensors-25-02544-f005:**
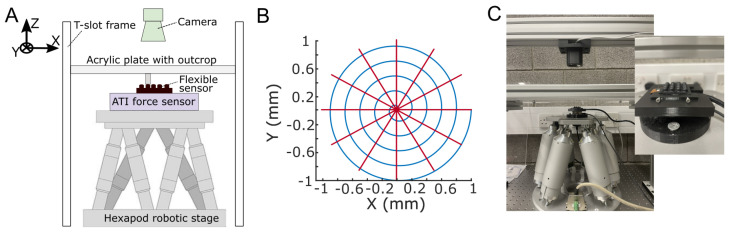
The robotic calibration platform. (**A**) Front-view illustration showing the force/torque sensor, carrying the Flexible LiVec sensor prototype, mounted on the hexapod robotic stage. Sensing elements of the prototype are brought into contact with a transparent outcrop screwed onto a rigid transparent acrylic plate. A camera mounted above the hexapod robotic stage captures the movement of the sensing element’s white tracking dot as an independent reference of XY displacement. (**B**) The XY displacement trajectories followed by the stage. These spiral and spoke patterns are repeated at steps of −0.1 mm of Z compression to a maximum Z compression of −1.5 mm. (**C**) Photo of the robotic calibration platform, including a closeup of the force/torque sensor and Flexible LiVec sensor prototype mounted on the hexapod.

**Figure 6 sensors-25-02544-f006:**
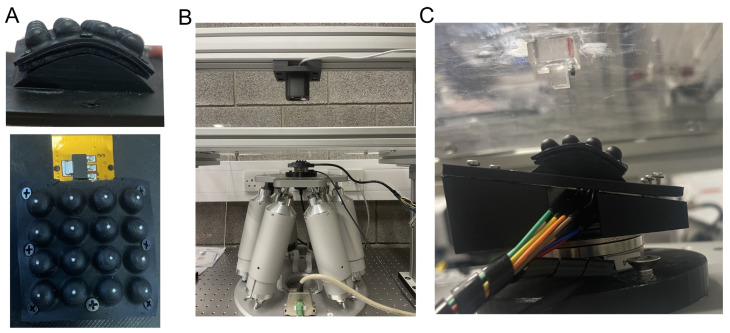
The flexed sensing validation setup, using the robotic calibration platform. (**A**) The flexible sensor is mounted in the 8 cm radius arc in a curved configuration. The top image is a side view, and the bottom image is a top view. These show how the relative positions of the sensing units change when the sensor is curved. (**B**) A front-view image of the robotic calibration platform for flexed sensing validation. (**C**) Front-view images of the flexible sensor on the robotic calibration platform showing the flexible sensor in the curved position, with the acrylic outcrop used for calibrating each individual sensing element above the sensor.

**Figure 7 sensors-25-02544-f007:**
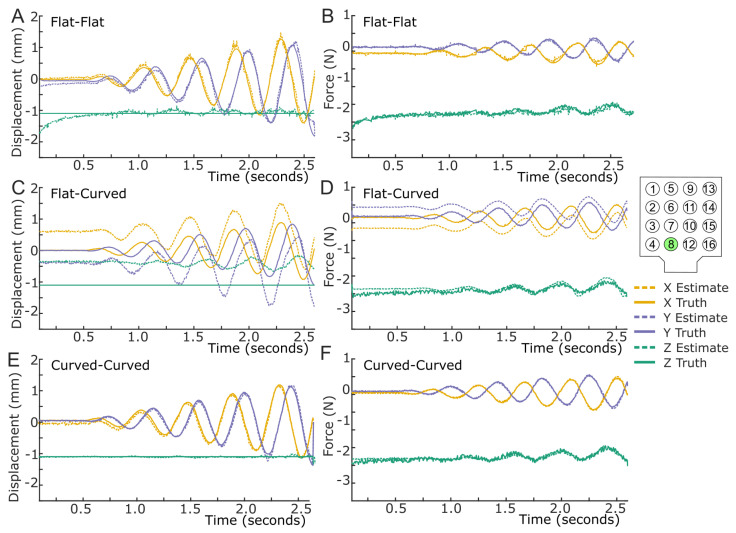
Displacement and force calibration testing results of sensing unit 8 (highlighted in the sensor schematic to the right of the graphs, chosen at random) for −1.1 mm of Z compression for one spiral calibration pattern. Actual (solid) and estimated (dashed) for the X (orange), Y (purple), and Z (green) axes. (**A**) Flat configuration calibration equations tested using test data collected in the flat configuration for displacement (referred to as Flat–Flat). (**B**) Flat configuration calibration equations tested using test data collected in the flat configuration for force (referred to as Flat–Flat). (**C**) Flat configuration calibration equations tested using test data collected in the curved configuration for displacement (referred to as Flat–Curved). (**D**) Flat configuration calibration equations tested using test data collected in the curved configuration for force (referred to as Flat–Curved). (**E**) Curved configuration calibration equations tested using test data collected in the curved configuration for displacement (referred to as Curved–Curved). (**F**) Curved configuration calibration equations tested using test data collected in the curved configuration for force (referred to as Curved–Curved).

**Figure 8 sensors-25-02544-f008:**
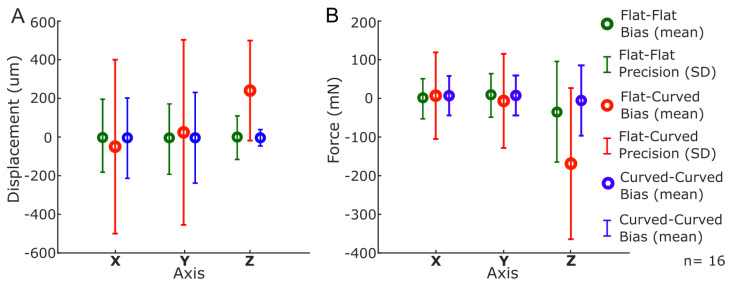
Error bar visualizations of the average bias and precision of all the sensing units (n = 16) for each axis, X, Y, and Z. (**A**) Displacement error bars for the three calibration configurations tested: Flat–Flat (green), Flat–Curved (red), and Curved–Curved (blue). (**B**) Force error bars for the three calibration configurations tested: Flat–Flat (green), Flat–Curved (red), and Curved–Curved (blue).

**Figure 9 sensors-25-02544-f009:**
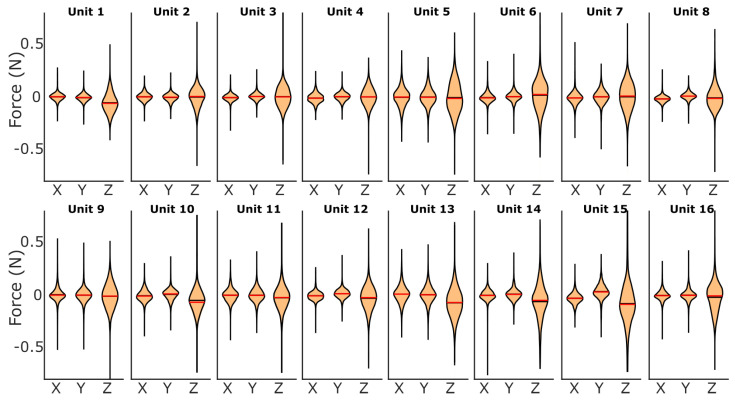
The X, Y, and Z force error distribution for each of the sixteen local sensing units, visualized in violin plots for the Flat–Flat configuration. The red line within the shaded area represents the mean error. The black line within the shaded area represents the median value. The black line tails represent the maximum positive and negative errors.

**Figure 10 sensors-25-02544-f010:**
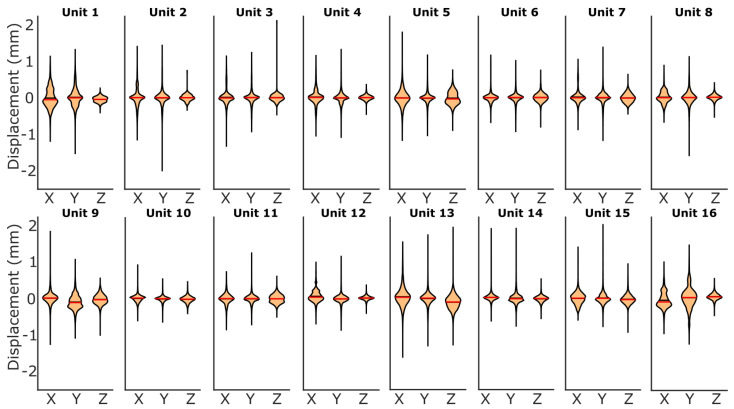
The X, Y, and Z displacement error distribution for each of the sixteen local sensing units, visualized in violin plots for the Flat–Flat configuration. The red line within the shaded area represents the mean error. The black line within the shaded area represents the median value. The black line tails represent the maximum positive and negative errors.

**Table 1 sensors-25-02544-t001:** Summary results for the bias and precision of force and displacement estimates across the sixteen sensing units for the three calibration configurations tested: (1) Flat–Flat, (2) Flat–Curved, and (3) Curved–Curved. The average bias (mean of sixteen bias values ($\bar{\mu }$)) of the sensing units and the average precision (mean of sixteen unit precision values ($\bar{SD}$)) for both the X, Y, and Z displacement (μm) and the X, Y, and Z force (mN).

			Sensor Unit Displacement		Sensor Unit Force	
**Train Dataset**	**Test Dataset**	**Axis**	**Mean Bias ($\bar{\mu }$) (μm)**	**Mean Precision ($\bar{\mathrm{SD}}$) (μm)**	**Mean Bias ($\bar{\mu }$) (mN)**	**Mean Precision ($\bar{\mathrm{SD}}$) (mN)**
		X	−3.62	180.89	−0.86	24.99
Flat	Flat	Y	3.83	172.26	2.09	53.56
		Z	−1.36	109.00	−2.78	135.33
		X	−59.63	449.97	6.93	112.02
Flat	Curved	Y	24.13	478.72	−6.65	121.93
		Z	248.57	258.90	−171.19	195.60
		X	1.58	221.61	1.04	54.49
Curved	Curved	Y	0.29	247.74	−0.32	55.16
		Z	−1.99	44.93	−1.31	97.15

**Table 2 sensors-25-02544-t002:** Comparison of the characteristics of flexible tactile sensors for robotic manipulation, which all use a flexible PCB. Not applicable is referred to as N/A. Information not available is marked as —.

Sensor	1. This work	2.	3. Omnitact	4. Digit360	5. uSkin	6.	7.	8.	9.	10.
**Transduction principle**	Optical	Optical	Optical–Camera	Optical–camera	Magnetic	Capacitive	Capacitive	Piezo-resistive	Piezo-resistive	Piezo-resistive
**Number of sensitive elements**	16	1	—	≈8.3 million	24	9	18	54	16	3
**Sensor shape**	Square grid	Square	Cylindrical	Cylindrical	Envelope fingertip	Square grid	Whole phalanx of finger	Hand shape	Square grid	Whole phalanx of finger
**Thickness (mm)**	7.2	2.0	33 ^†^	33 ^††^	6.05	1.4	4 ^†††^	—	—	—
**Robotic gripper integration**	N/A	N/A	Two custom-built 3D printed phalange grippers	Custom multi-finger robot hand	Allegro hand fingertip	ReFlex three-finger robot	i-limb robotic hand	Tactile glove	N/A	Humanoid robotic hand
**Measure-ments**	Three-axis force Three-axis displacement Vibration	Three-axis force	Normal force Shear force	Normal force Shear force Vibration	Three-axis force	Three-axis force	Three-axis force	Normalpressure	Normal pressure	Distributed pressure
**Force precision (mN)**	Fx: 25 Fy: 54 Fz: 135	1.6% ***	—	Fx/Fy: 1.27 Fz: 1.01	—	Fx: 0.471 * Fy: 0.466 * Fz: 0.201 *	Fx: 1.59 * Fy: 1.49 * Fz: 1.87 *	—	—	Fx/Fy: 3.4% Fz: 1.32% **
**Sensing range**	Fx/Fy:±1 N Fz: 0–35 N	1–10 N	—	Fx/Fy: <20 N Fz: <40 N	Fz: 7 N	Fx/Fy: 0.6 N Fz: 0.1–15 N	Fx/Fy:±1 N Fz: 0.1–3 N	1–500 kPa	0–70 kPa	Fx/Fy: 0–150 kPa Fz: 0–200 kPa
**Publication year**	2025	2021	2020	2024	2018	2019	2021	2015	2017	2020
**Reference**	—	[33]	[29]	[32]	[22]	[17]	[18]	[19]	[21]	[20]

^†^ This refers only to the height of the sensitive area and not the sensing components housing. ^††^ This is calculated from a base radius of 9 mm and a sensing area of 2340 mm^2^ and only refers the sensing area not including sensing component housing. ^†††^ This refers to the rubber bumps only, i.e., not including PCB. * These values correspond to force sensitivity, where sensitivity is in V/N. ** The units reported here are kPa^−1^. *** This is linear sensitivity with units of N^−1^.

## Data Availability

The data presented in this study are available upon request from the corresponding author.

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
