# Peer review of "On the Feasibility of Adapting the LiVec Tactile Sensing Principle to Non-Planar Surfaces: A Thin, Flexible Tactile Sensor"

_sensors, 2025, doi:10.3390/s25082544_

Round 1
Reviewer 1 Report
Comments and Suggestions for Authors
The authors propose a novel flexible tactile sensor based on the LiVec sensing principle, which is capable of providing localized 3D force and displacement measurements across a curved surface. The sensor utilizes a flexible PCB and soft components, eliminating the need for rigid circuit boards, and is designed to conform to non-planar surfaces such as robotic fingers and palms. This work presents a significant advancement in the field of tactile sensing for robotic manipulation. Considering that, the reviewer suggests that this work can be considered for publication after major revisions:
1. While the paper presents promising results, the experimental validation is limited to a single curvature (8 cm radius). To strengthen the claims, the authors should consider testing the sensor on multiple curvatures, which would provide a more comprehensive understanding of the sensor's performance in real-world applications.
2. In this manuscript, the authors do not mention Table 2, and the corresponding description should be added in the manuscript.
3. The authors should elaborate on how the calibration coefficients are affected by changes in curvature and whether there is a need for dynamic recalibration in real-time applications.
4. The introduction need to be rewritten.There are some important references that can be added. Some examples:
https://pubs.acs.org/doi/full/10.1021/acsnano.3c05838;
https://doi.org/10.1016/j.scib.2024.05.001;
https://doi.org/10.1002/advs.202408082.
1. It is not recommended to use the first person while writing. Eg: we
2. There are several minor mistakes that need to be corrected. Numbers and units should be separated by a space, such as “30×30mm2” in page 4, line 137. For better reading, the format should be standardized, such as “(Figure 5A and C)” in page 6, line 222 and “…in Figure 7E & 7F” in page13, line 398.
Author Response
Comments and Suggestions for Authors:
The authors propose a novel flexible tactile sensor based on the LiVec sensing principle, which is capable of providing localized 3D force and displacement measurements across a curved surface. The sensor utilizes a flexible PCB and soft components, eliminating the need for rigid circuit boards, and is designed to conform to non-planar surfaces such as robotic fingers and palms. This work presents a significant advancement in the field of tactile sensing for robotic manipulation. Considering that, the reviewer suggests that this work can be considered for publication after major revisions:
- While the paper presents promising results, the experimental validation is limited to a single curvature (8 cm radius). To strengthen the claims, the authors should consider testing the sensor on multiple curvatures, which would provide a more comprehensive understanding of the sensor's performance in real-world applications.
We certainly cannot disagree that testing on more curvatures could only make the results more comprehensive. Unfortunately, to be completely transparent, the project has now ended and the researcher leading this work (Olivia Leslie) is now working in industry full-time, so performing additional experimental work is currently not feasible; our hope was to publish the results of this proof-of-concept work with the current methodology. However, in this paper, with just two curvatures, we show that our current calibration method is not robust to changes in curvature. This is shown through the comparison of the bias and error of the sensor in the curved configuration using the calibrations performed in the flat and curved configurations. Additional experiments would not change this conclusion. Future designs that aim to remedy this weakness of the method must demonstrate its robustness across a range of curvatures, and ideally under dynamic flexion of the sensor backplane. I hope the reviewer can accept the benefits of sharing this work with the field, with methodological and performance limitations fully acknowledged. We have modified the text in the discussion to further clarify the limitations of this sensor.
Modified text in discussion section 4.5:
“4.5. Limitations and future considerations
The aim of the Flexible LiVec sensor was to further develop the LiVec sensor family by adapting the sensing principle to curved or flexible surfaces of robotic hands, such as the curved phalange or palm. However, it is important to note some limitations in the design and testing of the flexible LiVec sensor.
One limitation is that the current design evaluation of the Flexible LiVec sensor was only calibrated and then tested in three curvature configurations (Flat-Flat, Flat-Curved and Curved-curved). Expanding testing to include multiple curvatures would provide a more comprehensive understanding of the sensor’s performance across a broader range of real-world applications. However, additional curvature testing would not change the conclusion that our proposed calibration method, while offering significant benefits, is not robust to post-calibration changes in curvature. The results show that calibration coefficients determined in a flat (unflexed) position exhibit significant biases when applied to data acquired in a curved/flexed configuration. However, these biases are reduced when the sensor is both calibrated and tested in the same curved/flexed configuration. This highlights the importance of calibrating the sensor in its intended flexed configuration prior to use, and maintaining that curvature for a valid calibration.
The observed changes in calibration coefficients with varying curvature can be attributed to several factors. As the sensor flexes, the relative position of the reflector to the light-angle sensor may shift, altering the detected optical signal. Additionally, the light-angle sensor itself may become angled relative to the reflector, further affecting the measured light path. The shape of the internal cavity within the skin could also deform when the sensor is flexed, and the reflector may stretch, subtly changing its shape, all affecting the transmission, reflection, and reception of light relative to that observed in the curvature configuration for which the sensor was calibrated. All of these factors influence the optical path between the LEDs and light-angle sensor, introducing systematic errors in the calibration coefficients, resulting in observed measurement biases.
A potential solution to mitigate these biases is the implementation of dynamic recalibration or compensation techniques to maintain sensor accuracy. This could be achieved through real-time correction methods, such as adaptive calibration algorithms that update the calibration coefficients based on detected changes in sensor flexion. Alternatively, model-based compensation techniques could be employed to estimate and correct the expected distortions caused by flexion. However, any such method must be designed for real-time implementation, ensuring minimal computational delay to prevent performance lag. Future work should explore these approaches to improve sensor robustness under dynamic flexion and enhance its performance in robotic applications.
Beyond adapting the calibration coefficients, material and structural optimization could further enhance sensor performance. One solution could be adjusting the stiffness of the sensor’s skin, which would allow for a trade-off between force measurement sensitivity and measurement range. Softer materials could enhance sensitivity to small force changes, whereas stiffer materials would enable a broader force measurement range by resisting deformation. A different solution could mount the flexible sensor onto a curved but rigid surface to help mechanically stabilize the sensing units, minimizing unwanted deformation of the skin that could otherwise introduce variability in sensor readings. Additionally, directly glueing the sensor skin to the PCB could reduce the risk of blistering, improving the precision of the sensor. However, in line with the conclusions of the paper, methods that aim constrain the curvature of the sensor will require that the sensor is calibrated in this configuration prior to use.
The sensitivity of the sensor is also influenced by its physical size and layout, primarily due to the constraints imposed by the off-the-shelf light-angle sensor. The current square grid layout of the sensing units is approaching the maximum achievable density given the current design approach. To reduce the overall sensor size while maintaining performance, all component spatial dimensions would need to scale proportionally, to ensure that sensitivity is preserved. However, this would subsequently require enhancement of the sensitivity of the detector or an increase in the intensity of the LEDs to retain sufficient signal-to-noise ratio.
Another limitation is that the current evaluation did not assess the sensor’s durability or the stability of its calibration coefficients during continuous dynamic applications. While the previous version of the sensor (LiVec Finger [30]) was successfully demonstrated in a basic robotic manipulation task, the Flexible LiVec sensor has not yet been tested in robotic applications, but would be expected to perform similarly.
Previous work on the LiVec sensing principle [31] identified temperature-induced output drift as a limitation affecting measurement precision. However, this issue can be mitigated using a standard approach of regular rebiasing, a method commonly employed in other force/torque sensors [49– 52]. Future work will focus on characterizing the behavior of the LiVec sensing principle, more generally, in response to varying environmental conditions, including changes in temperature and humidity.
Despite the limitations of the Flexible LiVec sensor, it still demonstrates valuable multi-axis sensing capabilities, which might be sufficiently accurate for less precision-critical areas of a robotic hand, such as the palm or a phalanx.”
- In this manuscript, the authors do not mention Table 2, and the corresponding description should be added in the manuscript.
We apologize for not referencing Table 2 directly. The corresponding description has been added to the manuscript and the table is referred to throughout the discussion section.
Modified text in discussion section:
“The Flexible LiVec sensor improves miniaturization over other sensors in the LiVec sensor family, offering a thinner design and more densely packed sensing units that rival the most compact non-camera-based optical sensors available [10, 32 , 37]. The Flexible LiVec sensor is 40 % thinner than the LiVec finger, with an overall thickness from the tip of the sensing units to the back of the sensor casing of 7.2 mm. The compact arrangement of the sensing units in a uniform 4 × 4 grid design provides a higher spatial resolution than the LiVec Finger, with a shortest center-to-center distance between sensing units of 7 mm. Table 2 compares the characteristics of flexible tactile sensors, highlighting the wide measurement range and good precision of the flexible LiVec sensor. Table 2 also provides insight into the variety of shapes seen in flexible sensors designed for robotic fingers, typically specialised designs for a finger or a palm. The individual sensing unit design of the flexible LiVec means it can be designed in a modular configuration to instrument various robotic finger shapes."
- The authors should elaborate on how the calibration coefficients are affected by changes in curvature and whether there is a need for dynamic recalibration in real-time applications.
The results and conclusions in relation to this issue are presented in Section 4.3 (Measurement bias and precision), where curvature effects on the measurement bias and precision calibration coefficients are discussed. We have also expanded this discussion further to include a section on limitations discussing potential mechanisms which may directly influence the calibration coefficients.
Modified text in discussion section 4.5:
4.4 Measurement bias and precision
Flat testing
The Flexible LiVec sensor was tested in a flat (unflexed) position and showed a reduction in precision compared to the LiVec Finger [30]. The average force precision of the LiVec Finger (Table 1) is better across all axes, with average precision values of 20.89 mN, 19.19 mN, and 43.22 mN for the X, Y, and Z axes, respectively. In comparison, the Flexible LiVec sensor exhibits higher precision values of 24.99 mN, 53.56 mN, and 135.33 mN for the same axes (Table 1). Similarly, the displacement precision of the LiVec Finger is more consistent, with values of 56.70 μm, 50.18 μm and 13.83 μm for the X, Y, and Z axes.
The Flexible LiVec sensor, however, shows larger displacement average precision values, particularly in the X and Y axes, with 180.89 μm and 172.26 μm, respectively. Similarly, the average displacement precision of the LiVec Finger is more consistent, at values of 56.70 μm, 50.18 μm and 13.83 μm for the X, Y, and Z axes. The Flexible LiVec sensor, however, shows larger displacement precision values, particularly in the X and Y axes, with 180.89 μm and 172.26 μm, respectively. Although the bias is slightly smaller in some axes for the flexible sensor design, the overall precision of the LiVec Finger remains more reliable, particularly for applications requiring stable and accurate force and displacement measurements. These differences in bias and precision could be attributed to design variations between the sensors. The Flexible LiVec sensor’s soft, flexible back contrasts with the rigid back of the LiVec Finger, which likely provides less stability to the sensing elements and contributes to a less consistent zero point, especially in the Z-axis. Additionally, the absence of a top plate in the flexible design may allow more skin movement, leading to inaccuracies due to mechanical coupling from adjacent sensing units. In the previous LiVec sensor design, the top plate securely attached the deformable skin to the PCB, preventing blistering and ensuring the skin remained in place. In the flexible sensor, it was observed that compressing one sensing unit, particularly in the middle of the array, could cause slight movement in adjacent units, decreasing the sensor’s overall precision. The number of sensing units in the Flexible LiVec sensor was increased from 10 to 16; however, this change did not impact performance, as it was determined that the removal of the top plate to allow for flexibility in the design introduced the blistering, which was the cause of the negatively impacted precision.
Curved testing
To obtain accurate estimates of local force and displacement, calibration is required; however, for the Flexible LiVec sensor, the position of the sensor, either flat (unflexed) or curved (flexed), influenced the calibration outputs. When the sensor was calibrated in a flat configuration, but later tested in a curved configuration, significant biases in X and Y measurements for both displacement and force were observed (see Figure 7C and D).
The observed biases can likely be attributed to the shift in the reflector position relative to the light angle sensor when moving from a flat to a curved configuration and/or the light angle sensor being angled instead of normal to the reflector. Calibrating the sensor while it is in the curved configuration helps to account for these positional changes, thereby improving measurement accuracy. The sensor was configured with a larger curvature to assess this performance, which caused this bias shift. Therefore, flat calibration might perform better at smaller curvatures, but calibration in the actual configuration in which the sensor is deployed becomes crucial as the curvature increases.
4.5. Limitations and future considerations
See Comment 1.
- The introduction need to be rewritten. There are some important references that can be added. Some examples:
https://pubs.acs.org/doi/full/10.1021/acsnano.3c05838;
https://doi.org/10.1016/j.scib.2024.05.001;
https://doi.org/10.1002/advs.202408082.
Thank you for your suggestion. We have updated and rewritten the introduction and included the above references.
Modified text:
“1. Introduction
Mechanoreceptors are distributed throughout the human hand, with the highest concentration in the fingertips, enabling comprehensive tactile feedback, which is essential for dexterous manipulation [1, 2]. The natural curvature of the human fingertip and sensation across the skin of the fingers and palm enhances object control by inherently ensuring consistent surface contact, aiding in collision detection and re-grasping [1 ,3 ,4]. The curved shape of the finger also allows for better conformability to the surface of objects, enabling more stable and implicitly adaptive grips [3]. It has been shown that when humans manipulate objects, there is significant use of the curved sides of the fingers [ 5,6]. This curvature facilitates even force distribution, improving the reliability of the grasp and minimizing the risk of slippage or damage to the object [3].
Replicating human-like tactile capabilities in robotic systems requires comprehensive sensing across the entire surface of the end-effector. These sensors must provide high-precision, localized measurements of multi-axis force and deformation, particularly on non-planar, curved surfaces such as robotic phalanges and palms. Tactile feedback in these areas is crucial for achieving a secure and stable grip. However, most existing tactile sensors capable of measuring multi-axis force and deformation are flat or planar [7-10]. This design limitation fails to replicate the curved nature of human fingers and restricts the amount of tactile data that can be collected.
Flexible tactile sensors offer a promising alternative, as they can conform to the non-planar surfaces of robotic fingers, phalanges, and palms [11]. However, the complex curvature of robotic hands presents significant technical challenges for sensor design. Developing flexible tactile sensors that provide multi-axis force sensing and detailed slip information requires balancing flexibility, performance, and comprehensive data acquisition [3, 11]. As a result, many existing flexible sensors remain limited in their measurement capabilities. For instance, Ji et al. introduced a templated laser-induced graphene (TLIG)-based tactile sensor with high sensitivity, broad detection range, and excellent stability, making it well-suited for applications like wearable health monitoring and texture recognition [12] However, despite these strengths, the sensor is primarily limited to pressure measurement. Similarly, Yang et al. developed an iontronic pressure sensor using vertical graphene (VG) electrodes, achieving ultra-high sensitivity and mechanical stability alongside the ability to recognize different object shapes [13]. Another approach by Chen et al. utilized a biomimetic tactile sensing array with programmable micro-dome structures to enhance pressure sensitivity and spatial resolution [14]. While these methods are highly flexible and durable, they focus on pressure sensing, rather than capturing full tactile interactions, such as shear force, torque, or slip.
A common approach to flexible sensor design for robotic hands involves the use of flexible PCB technology, enabling tactile sensors to conform to curved surfaces and articulate at robotic joints [12, 13]. For example, Wang et al. developed a flexible 3 × 3 array of capacitive elements capable of measuring contact force and slip, designed to wrap around the curved surface of a human thumb [14]. A later iteration extended this design to robotic fingers, allowing for joint articulation [15]. While this capacitive approach enables thin and flexible sensors, it remains limited in force range (maximum of 3 N) and is prone to mechanical coupling interference and cross-talk between sensing units [15]. Similarly, piezoresistive sensors have been explored for flexible tactile sensing, such as Buscher et al.’s pressure-sensitive glove, which wraps around a robotic gripper [16]. Zhu et al. proposed another approach with three smaller piezoresistive sensors distributed across a robotic phalanx, allowing the sensor to flex at the joints [17]. However, while highly flexible, piezoresistive sensors primarily measure pressure and lack full multi-axis force sensing capabilities [16–18].
Another notable example of a curved robotic fingertip sensor is the uSkin sensor, which employs a magnetic sensing principle with an array of permanent magnets suspended above Hall effect MEMS sensors [19]. Its small, modular design allows it to be effectively integrated into robotic hands such as Allegro and iCub. However, magnetic sensors like uSkin are inherently vulnerable to external magnetic interference from commonplace sources such as ferromagnetic materials, which can compromise sensing accuracy.
In contrast, optical sensing techniques offer several advantages, including high sensitivity, immunity to electromagnetic interference, and the ability to perform multi-axis, multi-measurement sensing. Camera-based sensors are capable of providing high-resolution measurements and often feature non-flat, curved skins. Many incorporate hemispherical tips to mimic the human fingertip [7, 8, 20 –25], while others extend sensing capabilities along cylindrical finger surfaces [22, 26]. Among these, the GelSight Fin Ray sensor shows promise due to its ability to passively adapt to robotic grippers, providing high spatial resolution 3-axis force sensing and object orientation estimation [27, 28]. This is achieved by embedding mirrors and a camera within a fin ray effect gripper. A recent development is Meta’s Digit 360, a compact optical sensor approximately the size of a human fingertip. It is capable of measuring 3-axis forces in the milli-newton range while also detecting vibrations and spatial features with a resolution as fine as 7 microns [29]. This is an advancement in multidirectional camera-based optical sensing within a small, efficient form factor. However, its applicability remains constrained by its rigid structure, limiting its ability to conform to curved surfaces. Despite their advantages, camera-based sensors encounter several challenges in flexible sensing applications. Their inherent rigidity and bulkiness limits their ability to conform to highly curved or articulated surfaces.
Alternatively, optical fibres have been explored as a flexible optical tactile sensing solution due to their thin, lightweight, and highly adaptable nature [30]. Jiang et al. demonstrated the use of an optical microfiber-based slip sensor, approximately 2 mm thick, capable of measuring 3-axis force. However, despite incorporating fingerprint-mimicking ridges, this sensor is limited to a single measurement point over a relatively large sensing area of 2.5 cm x 7.5 cm.
The advantages of optical sensing for multi-axis tactile perception, combined with the limitations of existing optical approaches, showed the need for non-camera-based optical tactile sensors capable of instrumenting complex non-planar surfaces, such as robotic palms and fingers. In our previous work, the LiVec Finger was presented [31], which offers a thin, distributed multi-axis measurement sensor, using our novel non-camera-based optical sensing principle referred to as the LiVec principle [32]. However, the current LiVec Finger design is limited to tactile sensing over a planar sensor array area, requiring the object surface to be oriented parallel to the sensor surface (or vice versa) to ensure both a secure grasp and adequate tactile sensation regarding the contact interface. To allow robotic grippers to work more effectively in unstructured environments, having curved fingers with tactile sensing can reduce the need for perfectly normal contact between the sensor and the object. Therefore, having tactile sensation on all parts of a robotic gripper (fingers and palm) is beneficial. However, implementing the LiVec sensing principle to such non-planar surfaces requires us to adapt the sensor design.
Therefore, this paper presents a novel innovation of the LiVec sensor array concept: the Flexible LiVec sensor. This proof-of-concept design uses a flexible PCB and eliminates the rigid circuit board, creating a sensor capable of providing localized 3D force and 3D displacement measurements across an array of sensing units in a small-thickness, non-camera-based optical tactile sensor but across a curved surface. The aim is that the Flexible LiVec sensor will ultimately be able to conform to curved surfaces, such as the palms or joints of robotic hands, delivering precise, distributed tactile sensation across the entire robotic hand and beyond. The design concept combines the advantages of flexibility and a dense array of sensing units in a compact form, intending to make it well-suited for integration into the curved areas of existing robotic grippers.
The contribution of this paper is a demonstration of the feasibility of developing of a thin, flexible, non-camera-based optical tactile sensor that can conform to curved surfaces. The sensor presented here is a novel iteration of previous designs in the LiVec sensing family, now using flexible PCB technology to instrument non-planar surfaces. The flexible sensor is characterized in multiple calibration and testing configurations, demonstrating its capability for real-time measurements of distributed 3D displacement and 3D force and insights providing crucial insights on future calibration requirements of designs.
The rest of this paper is organized as follows. Section 2 describes the design and fabrication of the flexible skin-like sensor using the LiVec sensing principle. Additionally, the experimental calibration procedure is presented, which includes two configuration variations for investigation. Section 3 presents the results for the two calibration configuration variations. Section 4 compares the flexible skin-like LiVec sensor against the sensing capability and precision of the previous LiVec sensors [ 31 , 32 ] and to various state-of-the-art flexible tactile sensors. Section 5 draws conclusions and suggests future work.”
Comments on the Quality of English Language:
- It is not recommended to use the first person while writing. Eg: we
Thank you for the recommendation; we have addressed the use of first-person language throughout the text; please see the tracked changes manuscript.
- There are several minor mistakes that need to be corrected. Numbers and units should be separated by a space, such as “30×30mm2” in page 4, line 137. For better reading, the format should be standardized, such as “(Figure 5A and C)” in page 6, line 222 and “…in Figure 7E & 7F” in page13, line 398.
We apologize for the mistakes and have corrected all the above as well as further proofreading the manuscript for further mistakes. Please see the tracked changes version of the manuscript for the corrections.

Reviewer 2 Report
Comments and Suggestions for Authors Based on the context of the study, the following fivequestions are posed: 1. How does the sensor perform in actual robotic
operations, especially in terms of stability and durability during continuous dynamic applications? 2. Is the sensitivity and accuracy of the sensor
limited by its physical size and layout? If so, what
are the potential optimization schemes? 3. Has the research considered the effects of
environmental factors, such as changes in temperature
and humidity, on sensor performance? 4. What are the manufacturing costs and usability of
the sensor, especially in terms of feasibility for
mass production? 5. Is it possible to apply this sensing technology to
other types of sensors, such as pressure or temperature
sensors?
Author Response
Comments and Suggestions for Authors:
Based on the context of the study, the following five questions are posed:
- How does the sensor perform in actual robotic operations, especially in terms of stability and durability during continuous dynamic applications?
The sensor presented here was not tested in an actual robotic manipulation application. However, a previous version, the LiVec Finger [31], was successfully demonstrated in a robotic application. Based on these results, we expected that the flexible LiVec sensor will perform similarly to the LiVec Finger during robotic manipulation tasks. We chose to publish this work in MDPI Sensors due to the journal’s focus on novel sensor design methodologies, rather than application-focused research. Future work will focus on application-based studies, including demonstrating the sensor’s effectiveness for robotic gripping and quantitatively assessing its stability and durability in practical use. A discussion section of the paper, highlighting the limitations of the sensor and future work, has been added.
Modified text in discussion section 4.5:
“4.5. Limitations and future considerations
The aim of the Flexible LiVec sensor was to further develop the LiVec sensor family by adapting the sensing principle to curved or flexible surfaces of robotic hands, such as the curved phalange or palm. However, it is important to note some limitations in the design and testing of the flexible LiVec sensor.
One limitation is that the current design evaluation of the Flexible LiVec sensor was only calibrated and then tested in three curvature configurations (Flat-Flat, Flat-Curved and Curved-curved). Expanding testing to include multiple curvatures would provide a more comprehensive understanding of the sensor’s performance across a broader range of real-world applications. However, additional curvature testing would not change the conclusion that our proposed calibration method, while offering significant benefits, is not robust to post-calibration changes in curvature. The results show that calibration coefficients determined in a flat (unflexed) position exhibit significant biases when applied to data acquired in a curved/flexed configuration. However, these biases are reduced when the sensor is both calibrated and tested in the same curved/flexed configuration. This highlights the importance of calibrating the sensor in its intended flexed configuration prior to use, and maintaining that curvature for a valid calibration.
The observed changes in calibration coefficients with varying curvature can be attributed to several factors. As the sensor flexes, the relative position of the reflector to the light-angle sensor may shift, altering the detected optical signal. Additionally, the light-angle sensor itself may become angled relative to the reflector, further affecting the measured light path. The shape of the internal cavity within the skin could also deform when the sensor is flexed, and the reflector may stretch, subtly changing its shape, all affecting the transmission, reflection, and reception of light relative to that observed in the curvature configuration for which the sensor was calibrated. All of these factors influence the optical path between the LEDs and light-angle sensor, introducing systematic errors in the calibration coefficients, resulting in observed measurement biases.
A potential solution to mitigate these biases is the implementation of dynamic recalibration or compensation techniques to maintain sensor accuracy. This could be achieved through real-time correction methods, such as adaptive calibration algorithms that update the calibration coefficients based on detected changes in sensor flexion. Alternatively, model-based compensation techniques could be employed to estimate and correct the expected distortions caused by flexion. However, any such method must be designed for real-time implementation, ensuring minimal computational delay to prevent performance lag. Future work should explore these approaches to improve sensor robustness under dynamic flexion and enhance its performance in robotic applications.
Beyond adapting the calibration coefficients, material and structural optimization could further enhance sensor performance. One solution could be adjusting the stiffness of the sensor’s skin, which would allow for a trade-off between force measurement sensitivity and measurement range. Softer materials could enhance sensitivity to small force changes, whereas stiffer materials would enable a broader force measurement range by resisting deformation. A different solution could mount the flexible sensor onto a curved but rigid surface to help mechanically stabilize the sensing units, minimizing unwanted deformation of the skin that could otherwise introduce variability in sensor readings. Additionally, directly glueing the sensor skin to the PCB could reduce the risk of blistering, improving the precision of the sensor. However, in line with the conclusions of the paper, methods that aim constrain the curvature of the sensor will require that the sensor is calibrated in this configuration prior to use.
The sensitivity of the sensor is also influenced by its physical size and layout, primarily due to the constraints imposed by the off-the-shelf light-angle sensor. The current square grid layout of the sensing units is approaching the maximum achievable density given the current design approach. To reduce the overall sensor size while maintaining performance, all component spatial dimensions would need to scale proportionally, to ensure that sensitivity is preserved. However, this would subsequently require enhancement of the sensitivity of the detector or an increase in the intensity of the LEDs to retain sufficient signal-to-noise ratio.
Another limitation is that the current evaluation did not assess the sensor’s durability or the stability of its calibration coefficients during continuous dynamic applications. While the previous version of the sensor (LiVec Finger [30]) was successfully demonstrated in a basic robotic manipulation task, the Flexible LiVec sensor has not yet been tested in robotic applications, but would be expected to perform similarly.
Previous work on the LiVec sensing principle [31] identified temperature-induced output drift as a limitation affecting measurement precision. However, this issue can be mitigated using a standard approach of regular rebiasing, a method commonly employed in other force/torque sensors [49– 52]. Future work will focus on characterizing the behavior of the LiVec sensing principle, more generally, in response to varying environmental conditions, including changes in temperature and humidity.
Despite the limitations of the Flexible LiVec sensor, it still demonstrates valuable multi-axis sensing capabilities, which might be sufficiently accurate for less precision-critical areas of a robotic hand, such as the palm or a phalanx..”
- Is the sensitivity and accuracy of the sensor limited by its physical size and layout? If so, what are the potential optimization schemes?
Thank you for your questions. The sensitivity and accuracy of the sensor are limited by the current size of the off-the-shelf light-angle sensor. The Flexible LiVec sensor design has a uniform sensing unit layout at the maximum density achievable with these components in the current design paradigm (which has many of the electronic components on the sensing board). To reduce the overall size while maintaining the sensitivity, it would be necessary for all the component spatial dimensions to scale down. However, the sensitivity of the detector or the intensity of the LEDs would need to increase to stay above the detection threshold of the light angle sensor, maintaining a good signal-to-noise ratio. To change the measurement sensitivity, material stiffness can always be changed to trade force measurement sensitivity for force measurement range, as this property of the skin is what defines the relationship between forces experienced and displacements experienced by the reflector. A discussion on these considerations has been added to the paper.
Modified text in methods section 2.0.1:
“The Flexible LiVec sensor consists of 16 sensing units arranged in a uniform 4 × 4 grid shown in Figure 1. Each sensing unit can estimate 3D point forces and displacements. The assembled sensor has an overall size of 30 mm × 30 mm × 7.2 mm, length, width and depth, respectively (Figure 1C and D). The 16 sensing units have a 7 mm pitch between units, a 6 mm external diameter at the base and a maximum height of 5 mm. This design varies from the LiVec Finger presented in [30] shown in Figure 2 by having:
- A 14% decrease in center-to-center sensing unit density (7 mm vs 8.15 mm).
- A 40% reduction in thickness (7.2 mm vs 12 mm).
- A more compact design, containing 60% more sensing units (dimensions: 30 × 30 mm2 vs 38.22 × 26.44 mm2).
- A removal of the rigid top plate and backbone assembly components to allow the sensor to be flexible.
- A flexible PCB.
The Flexible LiVec sensor uses the LiVec sensing principle [31]. Each local sensing unit consists of a deformable, cylindrical protrusion encompassing the local sensing components (see Figure 3). As each sensing unit is individual, this means that future designs can be made modular, allowing the arrangement of the sensing units to be specific to robotic phalanges or palms. The size of the flexible LiVec sensor is primarily constrained by the dimensions of its individual sensing units, which are determined by the off-the-shelf light-angle sensor and photometric front-end components. These factors influence both the overall sensor footprint and the density of sensing units in a uniform layout.
The LiVec sensing principle is presented in [30] in detail; the following is a brief summary. Each LiVec sensing unit infers XYZ force and displacement by measuring light reflected off an internal reflector within the skin protrusion. When an external force deforms the skin protrusion, this shifts the internal reflector’s position (and possibly orientation), altering the reflected light's direction and intensity. The light angle sensor captures this light and outputs four photocurrents, encoding light ray angles of arrival and intensities. Each sensing unit undergoes a calibration procedure to convert the photocurrents into XYZ force and displacement estimates. The vector sum of the local XYZ forces across all sensing elements is used to estimate the global force of the whole sensor array.”
Modified text in discussion section 4.5:
See Comment 1
- Has the research considered the effects of environmental factors, such as changes in temperature and humidity, on sensor performance?
This work primarily focused on adapting the LiVec sensing principle to a flexible design and assessing its feasibility for covering curved surfaces. As a result, the effects of environmental factors such as temperature and humidity were not directly investigated. However, previous studies on the LiVec sensing principle [30] have shown that external temperature fluctuations can influence the precision of force and displacement measurements. This issue can be mitigated through regular rebiasing, a common practice in other sensors, such as ATI force/torque sensors. The effects of humidity on the LiVec sensor have not yet been comprehensively explored, but potential impacts could arise. Future work will investigate these factors to better understand sensor performance in environments with changing environmental conditions.
Modification of text in the discussion section 4.5:
See comment 1
- What are the manufacturing costs and usability of the sensor, especially in terms of feasibility for mass production?
It is important to mention first that, as this paper reports a proof-of-principle design, this design is not intentionally optimized for mass production. And, of course, in the volumes we are manufacturing our prototypes, there are no bulk-buying cost benefits. The sensor utilizes off-the-shelf components and outsourced manufacturing and assembly processes (without discount). The estimated overall manufacturing cost of the flexible sensor is approximately US$ 1,050, which includes US$ 1,000 for producing the flexible PCB, US$ 25 for the Teensy 4.1 readout microcontroller, and US$ 25 for consumables required to fabricate the silicone sensor skins. Labour costs for skin manufacture and final assembly are not included. The most expensive components in the design are the light angle sensor and the photometric front-end chip (i.e., amplifier and data acquisition). Of course, given the modularity of the design and its ability to be arrayed, the final cost will also depend on the preferred number of sensing elements in the final design.
Modification of text in Methods section 2.0.1.
“The Flexible LiVec sensor uses the LiVec sensing principle [31]. Each local sensing unit consists of a deformable, cylindrical protrusion encompassing the local sensing components (see Figure 3). As each sensing unit is individual, this means that future designs can be made modular, allowing the arrangement of the sensing units to be specific to robotic phalanges or plans.”
- Is it possible to apply this sensing technology to other types of sensors, such as pressure or temperature sensors?
Thank you for your interesting question. Certainly, pressure measurement is entirely feasible; currently, pressure distributions could be estimated by interpolating the estimated normal force over the estimated sensing element XY positions and scaling per unit of XY area. Note, the resolution of these estimates would be limited by the spatial density of the sensing elements in the array. However, applying this to temperature sensing would be more complex, but not impossible, given the sensitivity of the optoelectronics to temperature variation; although existing solutions (for example, using thermistors) for direct temperature measurement may be more sensitive and practical.
